# Removal of Arsenic from Wastewater by Using Nano Fe_3_O_4_/Zinc Organic Frameworks

**DOI:** 10.3390/ijerph191710897

**Published:** 2022-09-01

**Authors:** Xuexia Huang, Yun Liu, Xinyi Wang, Linwei Zeng, Tangfu Xiao, Dinggui Luo, Jia Jiang, Hongguo Zhang, Yuhui Huang, Mingzhen Ye, Lei Huang

**Affiliations:** 1School of Environmental Science and Engineering, Guangzhou University, Guangzhou 510006, China; 2Key Laboratory for Water Quality and Conservation of the Pearl River Delta, Ministry of Education, Guangzhou University, Guangzhou 510006, China; 3Linkoping University-Guangzhou University Research Center on Urban Sustainable Development, Guangzhou University, Guangzhou 510006, China

**Keywords:** Fe_3_O_4_@ZIF-8, As(V), nano Fe_3_O_4_, adsorption properties

## Abstract

Efficient removal of arsenic in wastewater is of fundamental importance due to the increasingly severe arsenic pollution. In this study, a new composite adsorbent (Fe_3_O_4_@ZIF-8) for As(V) removal from wastewater was synthesized by encapsulating magnetic Fe_3_O_4_ nanoparticles into metal organic frameworks. In order to evaluate the feasibility of Fe_3_O_4_@ZIF-8 as an adsorbent for As(V) removal, the adsorption properties of Fe_3_O_4_@ZIF-8 were systematically explored by studying the effects of dosage, pH, adsorption isotherm, kinetics, and thermodynamics. Additionally, the characterization of Fe_3_O_4_@ZIF-8 before and after adsorption was analyzed thoroughly using various tests including SEM-EDS, XPS, BET, XRD, TG, FTIR, and the properties and arsenic removal mechanism of the Fe_3_O_4_@ZIF-8 were further studied. The results showed that the Fe_3_O_4_@ZIF-8 has a specific surface area of 316 m^2^/g and has excellent adsorption performance. At 25 °C, the initial concentration of arsenic was 46.916 mg/L, and pH 3 was the optimum condition for the Fe_3_O_4_@ZIF-8 to adsorb arsenic. When the dosage of the Fe_3_O_4_@ZIF-8 was 0.60 g/L, the adsorption of arsenic by the Fe_3_O_4_@ZIF-8 can reach 76 mg/g, and the removal rate can reach 97.20%. The adsorption process of arsenic to the Fe_3_O_4_@ZIF-8 can be well described by the Langmuir isotherm model and the second-order kinetic equation. At pH 3 and temperature 298 K, the maximum adsorption capacity of arsenic by the Fe_3_O_4_@ZIF-8 was 116.114 mg/g. Through the analysis of thermodynamic parameters, it is proved that the adsorption process of arsenic by the Fe_3_O_4_@ZIF-8 is a spontaneous endothermic reaction. The Fe_3_O_4_@ZIF-8 has broad prospects for removing As(V) pollution in wastewater, because of its strong adsorption capacity, good water stability, and easy preparation.

## 1. Introduction

Human survival and development are inseparable from water resources. With the continuous expansion of human productivity and the continuous development of the manufacturing industry, the demand for mineral resources continues to increase, and the generated pollutants enter the water body, resulting in more and more serious water pollution [1]. Arsenic is a non-metallic element, which widely exists in nature [2]. At present, hundreds of arsenic minerals have been found. Arsenic and its compounds are used in pesticides, herbicides, insecticides, and various alloys [3,4]. In 2017, the World Health Organization’s International Agency for Research on Cancer placed arsenic and inorganic arsenic compounds on the list of Class I carcinogens [5]. In 2019, arsenic and arsenic compounds were included in the “List of Toxic and Harmful Water Pollutants (First Batch)” by the Ministry of Ecology and Environment. The sources of arsenic pollution in water bodies are wastewater, waste gas and industrial waste discharged from industrial production, various organic arsenic pesticides used in agricultural production to avoid the impact of insect pests on crops, and the natural environment accompanied by volcanic eruptions and rock weathering and diffusion [6]. Arsenic-containing substances and arsenic can denature proteins and enzymes in cells, and cause cell damage by increasing reactive oxygen species in cells [7]. Studies have shown that the prevalence of arsenic poisoning is as high as 133‰ in groups who drink water with an arsenic content of 0.3 mg/L for a long time, and among patients with chronic arsenic poisoning, the cancer rate is as high as 15% [8]. In the World Health Organization’s “Water Quality Standards for Drinking Water”, the index value of arsenic is 0.01 mg/L. Arsenic pollution in water will not only cause serious harm to the human body, but also harm the normal growth of plants, affecting the operation of internal water and chlorophyll [9]. Therefore, arsenic pollution in water has become an urgent problem to be solved. The arsenic removal technologies reported in the literature include oxidation, coagulation/filtration, ion exchange, membrane method, biological method, electro-flocculation, and adsorption method [10,11]. The adsorption method has simple steps and is more feasible for practical application [12,13]. MOF (Metal-organic framework) materials can effectively provide tunable porosity [14], stable pore structure [15], geometry, chemical function [16], structural uniformity, and ultra–high surface area [17].

MOFs are adsorbents composed of porous crystalline materials with ultra−high surface area and highly ordered structures composed of organic ligands and metal ions linked in a three-dimensional lattice, which are promising in the fabrication of various attractive new compounds [18]. It was reported that the first MOF material for the removal of arsenic contamination from water was Fe-BTC. Fe-BTC, consisting of iron nodes and 1,3,5 trimesic acid groups, was synthesized by a facile solvothermal method and used as an adsorbent to adsorb As(V) in the pH range of 2–10. At pH 4; the adsorption capacity of Fe-BTC was 12.3 mg/g, which was more than six times that of Fe_2_O_3_ nanoparticles [19]. By infrared measurement analysis, a new Fe-OAs group in the infrared band appeared at 824 cm^−1^, which confirmed the adsorption of As(V). In addition, by transmission electron microscopy, it was confirmed that the adsorption of As(V) occurred at the interior of the MOFs and interacts with the Fe nodes of the framework. Although the adsorption capacity is not strong, this is a landmark discovery in the research on arsenic removal by MOFs, which indicates that MOFs materials have great potential for arsenic capture. Moreover, the researchers also conducted studies on the removal of arsenic by classical MOFs. The reported adsorption capacity of ZIF-8 for arsenic at low equilibrium concentration (9.8 µg/L) was 76.5 mg/g. The study of the adsorption mechanism showed that the adsorption was due to the dissociative adsorption of water, which produced a large number of external active sites (Zn-OH) and formed an inner spherical complex with arsenate [20]. Due to its potential for arsenic adsorption, the research on the adsorption capacity of arsenic in wastewater provides a data basis for the treatment of arsenic pollution in wastewater.

The main objectives of this study were to: (1) synthesize a new composite adsorbent (Fe_3_O_4_@ZIF-8) by encapsulating magnetic Fe_3_O_4_ nanoparticles into metal organic frameworks for As(V) removal from wastewater, (2) evaluate the feasibility of Fe_3_O_4_@ZIF-8 as an adsorbent for As(V) removal by studying the effects of dos-age, pH, adsorption isotherm, kinetics, and thermodynamics, (3) analyze the characterization of Fe_3_O_4_@ZIF-8 before and after adsorption thoroughly using various tests including SEM-EDS, XPS, BET, XRD, TG, FTIR, and (4) study the properties and arsenic removal mechanism of the Fe_3_O_4_@ZIF-8.

## 2. Experimental Section

### 2.1. Chemicals

Arsenic standard solution (Beijing, China) was purchased from Beijing Northern Weiye Metrology Technology Research Institute. NH_4_OH (Tianjin, China) was purchased from Tianjin Damao Chemical Reagent Factory. FeCl_3_·6H_2_O and CH_3_OH (Tianjin, China) were purchased from Tianjin Zhiyuan Chemical Reagent Co., Ltd. In addition, FeCl_2_·4H_2_O (Shanghai, China), 2-methy imidazole (C_4_H_6_N_2_,Shanghai, China) were purchased from Shanghai McLean Biochemical Technology Co., Ltd. Zn(NO_3_)_2_·6H_2_O, and HNO_3_ (Guangzhou, China) were purchased from Guangzhou Chemical Reagent Factory. Deionized water is used throughout this study. The conductivity of H_2_O was less than 2 μs/cm. As(V) stock solution was prepared with 1001 μg/mL As(V) standard solution (H_3_AsO_4_, Beijing). The inductively coupled plasma atomic emission spectroscope (ICP-AES) was used to measure the As(V) concentration in the aqueous solutions.

### 2.2. Synthesis of Adsorbents

#### 2.2.1. Synthesis of Magnetic Fe_3_O_4_ Nanoparticles

Magnetic Fe_3_O_4_ nanoparticles were synthesized using a modified iron (II) and iron (III) co-precipitation method as described in the literature [21]. FeCl_2_·4H_2_O (10 mmol) and FeCl_3_·6H_2_O (10 mmol) were gradually poured into 30 mL deionized water and mixed by mechanical stirring, the solution was heated to 60 °C for 30 min in the ultrasonic cleaner. Then 40.00 mL of 3.50 mol/L ammonia solution was introduced into the mixed iron(II)/iron(III) solution with continuous stirring at 60 °C for 30 min. After the reaction was completed, the produced mixture was filtered and washed with deionized water until neutral and finally dried at 60 °C under the vacuum for 12 h to obtain Fe_3_O_4_ nanoparticles.

#### 2.2.2. Synthesis of Fe_3_O_4_@ZIF-8

The preparation of Fe_3_O_4_@ZIF-8 was improved according to reference [22]. Weighed 3.30 g of 2-methy imidazole and 0.10 g of magnetic nanometer Fe_3_O_4_, dissolved in 70.00 mL of methanol. The mixture was ultrasonicated for 15 min, 1.50 g of Zn(NO_3_)_2_·6H_2_O was weighed and dissolved in 70.00 mL of methanol, which was slowly added dropwise to the above-mentioned sonicated solution, mechanically stirred for 24 h, and then centrifuged. The obtained solid was washed with methanol at least three times, and then vacuum-dried at 80 °C for 12 h.

## 3. Results and Discussion

### 3.1. Adsorption Experiments

#### 3.1.1. Effect of Adsorbent Dosage

Prepared 46.916 mg/L As(V) solution, added 25 mL to several centrifuge tubes, and added Fe_3_O_4_@ZIF-8 adsorbent 5, 10, 15, 20, 25, 30, 35 mg respectively, at pH 3, temperature was 25 °C shook in a shaker for 24 h. The adsorption capacity and removal rate of As(V) by Fe_3_O_4_@ZIF-8 with the dosage of adsorbent are shown in Figure 1.

As illustrated in Figure 1, the equilibrium adsorption capacity of Fe_3_O_4_@ZIF-8 for As(V) shows decreasing trend with the dosage. It was because that the increase in the dosage of Fe_3_O_4_@ZIF-8, that the number of adsorption sites in water also increased, but the adsorption sites on the surface of Fe_3_O_4_@ZIF-8 had not reached saturation. Therefore, although the amount of adsorbed As(V) increased, the degree of saturation decreased, and Fe_3_O_4_@ZIF-8 the equilibrium adsorption capacity of As(V) decreased.

It can be seen in Figure 1 that when the dosage of Fe_3_O_4_@ZIF-8 was 10 mg, the removal rate of As(V) reached 76.29%. When the dosage of Fe_3_O_4_@ZIF-8 was increased to 15 mg, the removal rate of As(V) reached 97.20%. When the dosage of Fe_3_O_4_@ZIF-8 was 20–35 mg, the removal rate of As(V) reached more than 99%, and the removal was basically complete. Continue to increase the dosage of Fe_3_O_4_@ZIF-8 to As(V) the removal rate did not improve much. Therefore, from the perspective of economy and resource saving, the optimal dosage of Fe_3_O_4_@ZIF-8 in this experiment is 15 mg/25 mL, which is 0.6 g/L.

#### 3.1.2. Effect of pH

Prepared 25 mg/L As(V) solution, adjusted the pH from 2 to 11 by NaOH solution and HNO_3_ solution, pipetted 25 mL into several centrifuge tubes, and dosage of Fe_3_O_4_@ZIF-8 adsorbent was 10 mg. Shook on a shaker for 24 h at 25 °C. The adsorption capacity and removal rate of As(V) by Fe_3_O_4_@ZIF-8 are shown in Figure 2.

From Figure 2a, it can be found that the adsorption amount of Fe_3_O_4_@ZIF-8 reached the maximum when pH = 3.0, and the removal rate of As(V) was the largest; when pH < 3.0, the adsorption of Fe_3_O_4_@ZIF-8 decreased rapidly. The amount of As(V) adsorption by Fe_3_O_4_@ZIF-8 decreased gradually with the increase of pH when pH > 3.0. When pH > 9.0, the adsorption capacity and removal rate of As(V) decreased rapidly. This phenomenon can be explained by Figure 2b. The experimentally measured isoelectric point of Fe_3_O_4_@ZIF-8 was 9.46. Therefore, when pH < 9.46, the surface of Fe_3_O_4_@ZIF-8 was positively charged. According to the literature, when pH < 2, As(V) mainly exists in the form of H_3_AsO_4_ in water. When pH is 2–7, As(V) mainly exists in the form of H_2_AsO_4_^−^. When pH > 7, As(V) It mainly exists in the form of H_2_AsO_4_^−^ [23,24]. Combined with the experimental data, it can be seen that when pH = 2, the content of H_3_AsO_4_ was relatively high, which was not conducive to the adsorption of Fe_3_O_4_@ZIF-8. When pH = 3–9, the positive charge of Fe_3_O_4_@ZIF-8 decreased with the increase of pH and adsorption capacity for H_2_AsO_4_^−^ and HAsO_4_^2^^−^ decreased. When pH > 9.46, the positive charge on the surface of Fe_3_O_4_@ZIF-8 changed to negative charge, which repelled each other with HAsO_4_, so the adsorption amount and removal rate of As(V) reduced rapidly.

#### 3.1.3. Adsorption Isotherm

Prepared 25 mL of As(V) solution with an initial concentration of 40–120 mg/L and pH = 3 and added 10 mg of Fe_3_O_4_@ZIF-8 adsorbent. Shook for 24 h on a shaker at 25 °C, 35 °C and 45 °C. The obtained adsorption isotherms are shown in Figure 3. At different temperatures, the adsorption isotherms of Fe_3_O_4_@ZIF-8 for As(V) were fitted by Langmuir and Freundlich models as shown in Figure 3. From the data in Table 1, it can be seen that the Langmuir adsorption isotherm equation was more suitable for describing the adsorption process of As(V) by Fe_3_O_4_@ZIF-8, and the calculated correlation coefficient was higher than that of the Freundlich adsorption isotherm equation.

This indicated that the adsorption of As(V) by Fe_3_O_4_@ZIF-8 was monolayer adsorption. Moreover, the R_L_ value was between 0 and 1, which meant that the adsorption was easy to proceed. When the reaction temperature was 318 K, the maximum adsorption capacity of Fe_3_O_4_@ZIF-8 simulated by the Langmuir adsorption isotherm model was 125.628 mg/g. It can be found that Fe_3_O_4_@ZIF-8 has a good adsorption effect on As(V).

#### 3.1.4. Kinetic Study

A 250 mL of As(V) solution with a concentration of 30 and 50 mg/L and a pH of 3 was prepared. Then, 0.1 g of Fe_3_O_4_@ZIF-8 adsorbent was added and shaken at 25 °C in a constant temperature shaker. The fitted kinetic equation of As(V) adsorption by Fe_3_O_4_@ZIF-8 is shown in Figure 4.

It can be seen in Figure 4 that the fitting effect of the second-order kinetic equation for the adsorption of As(V) to Fe_3_O_4_@ZIF-8 was obviously better than that of the first-order kinetic model. From the data in Table 2, the correlation coefficient R^2^ of the second-order kinetic model was much larger than that of the first-order kinetic model, and both were higher than 0.999. Under different concentrations of As(V) solution, the equilibrium adsorption capacity of Fe_3_O_4_@ZIF-8 calculated by the second-order kinetic model was basically consistent with actual adsorption capacity. Therefore, the adsorption process of As(V) by Fe_3_O_4_@ZIF-8 satisfies the second-order kinetic model. This means that the adsorption of As(V) by Fe_3_O_4_@ZIF-8 is mainly a chemical adsorption process.

#### 3.1.5. Thermodynamic Studies

The thermodynamic parameters of Fe_3_O_4_@ZIF-8 adsorption of As(V) at temperatures of 298 K, 308 K, and 318 K were calculated by formula, as shown in Table 3.

It can be seen in Table 3 that the standard Gibbs free energy ΔG0 of Fe_3_O_4_@ZIF-8 adsorbing As(V) is less than 0 at different temperatures, which indicates that Fe_3_O_4_@ZIF-8 adsorbing As(V) is a spontaneous reaction, and with the increase of temperature, the absolute value of ΔG^0^ increases, which indicates that the increase of temperature plays a role in promoting the adsorption reaction. In addition, the standard enthalpy changes ΔH^0^ during the adsorption of As(V) by Fe_3_O_4_@ZIF-8 is greater than 0, indicating that the reaction is an endothermic reaction. The standard entropy change of Fe_3_O_4_@ZIF-8 adsorption of As(V) at different temperatures is ΔS^0^ > 0, which indicates that the system disorder becomes larger after the adsorption reaction, which is an entropy-driven process. The above data can prove that the adsorption process of As(V) by Fe_3_O_4_@ZIF-8 is a spontaneous endothermic reaction.

### 3.2. Characterization

#### 3.2.1. Scanning Electron Microscopy (SEM)

From a–c in Figure 5, it can be seen that the prepared Fe_3_O_4_@ZIF-8 has irregular columnar, spherical and cubic structure particles with rough surfaces. It can be seen from the figure that the particles were adhered to each other and stacked to form many channels, and this agglomeration phenomenon provided more adsorption sites for Fe_3_O_4_@ZIF-8 to adsorb arsenic. From the SEM image, it can be known that the particle size of the prepared Fe_3_O_4_@ZIF-8 was about 100 nm. It can be seen from pictures d–f in Figure 5 that the prepared Fe_3_O_4_ was a nanomaterial, which was approximately spherical particles, and the particles adhere to each other and agglomerate together. The particle size of the prepared magnetic nano Fe_3_O_4_ was much smaller than that of the Fe_3_O_4_@ZIF-8 material.

From Figure 6, it can be seen that the Fe_3_O_4_@ZIF-8 particle gap after adsorption of arsenic is reduced, which indicates that the arsenic in the water is adsorbed to the surface of Fe_3_O_4_@ZIF-8, thus filling the pores.

It can be seen from Figure 6 that the Fe_3_O_4_@ZIF-8 particle agglomeration phenomenon after adsorption is more obvious, which reflects the strong adsorption capacity of the material. The particle shape of Fe_3_O_4_@ZIF-8 after adsorption was changed and the particle size became smaller than that before adsorption.

#### 3.2.2. EDS Analysis

The EDS analysis of Fe_3_O_4_@ZIF-8 before and after arsenic adsorption are shown in Figure 7 and Table 4. Through EDS analysis of Fe_3_O_4_@ZIF-8, we can find that As element was not detected in Fe_3_O_4_@ZIF-8 before adsorption, while As element was detected in Fe_3_O_4_@ZIF-8 after adsorption. This proved that Fe_3_O_4_@ZIF-8 has adsorption capacity for arsenic. The Fe_3_O_4_@ZIF-8 before adsorption detected C, N, O, Fe and Zn elements, which were consistent with the elemental composition of the Fe_3_O_4_@ZIF-8 material, which proved that the metal-organic framework composite was successfully synthesized [25].

#### 3.2.3. XRD Analysis

Figure 8 was the XRD pattern of Fe_3_O_4_ and Fe_3_O_4_@ZIF-8. These include the XRD patterns of Fe_3_O_4_@ZIF8 before and after adsorption, the XRD patterns of the prepared magnetic nano Fe_3_O_4_, the corresponding ZIF-8 standard pattern [26], and the Fe_3_O_4_ standard card (JCPDS 19-0629).

Comparing the prepared magnetic nano Fe_3_O_4_ and Fe_3_O_4_ standard cards, it is found that the characteristic peak positions of the standard cards were the same, which can prove the successful preparation of magnetic nano Fe_3_O_4_. By comparing the standard cards of Fe_3_O_4_@ZIF-8, ZIF-8, and Fe_3_O_4_, it can be found that the characteristic peaks are basically the same as those of Fe_3_O_4_ and standard cards, and there were diffraction peaks consistent with the standard pattern of ZIF-8, which can prove that Fe_3_O_4_@ZIF-8 synthesis was relatively successful. Comparing the XRD patterns of Fe_3_O_4_@ZIF-8 before and after adsorption, it can be found that the position of the diffraction peak of Fe_3_O_4_@ZIF-8 does not change after adsorption of arsenic, and the relative intensity of the peak increases, which indicates that Fe_3_O_4_@ZIF-8 is adsorbing arsenic. The arsenic process did not destroy the structure and properties. Therefore, the material can exist stably in an aqueous solution.

#### 3.2.4. Thermogravimetric Analysis (TG)

In order to evaluate the stability of Fe_3_O_4_@ZIF-8 material, Fe_3_O_4_@ZIF-8 was heated from 30 °C to 800 °C under N_2_, and the stage of material weight loss was analyzed. As shown in Figure 9, it can be observed that Fe_3_O_4_@ZIF-8 was in a state of gradual weight loss at 30–264.5 °C, with a weight loss of 8.5%. This can be attributed to desorption of adsorbed water or solvent. At 264.5–800 °C, the weight loss rate of Fe_3_O_4_@ZIF-8 increased and remained in a weightless state, which may be due to the decomposition of the ligand ZIF-8. Comparing the weight loss changes of Fe_3_O_4_@ZIF-8 before and after adsorption, Fe_3_O_4_@ZIF-8 after adsorption of arsenic was heated from 30 °C to 516.5 °C, and the weight loss was 15.2%. It was obvious from Figure 9 that Fe_3_O_4_ after the adsorption of arsenic, the weight loss rate of @ZIF-8 was significantly slowed down. At 800 °C, the weight loss of Fe_3_O_4_@ZIF-8 before adsorption reached 66.4%, and the weight loss of Fe_3_O_4_@ZIF-8 after adsorption was 36.8%. This shows that Fe_3_O_4_@ZIF-8 has good thermal stability after adsorbing arsenic, and arsenic is adsorbed to the surface of Fe_3_O_4_@ZIF-8, and it is not easy to desorb and decompose.

#### 3.2.5. FTIR Analysis

From the infrared spectrum of magnetic nano Fe_3_O_4_ in Figure 10, it can be seen that absorption bands corresponding to the carboxylate groups on the surface of Fe_3_O_4_ NPs appeared at 1384.25 cm^−1^ and 1628.70 cm^−1^ [27]. At 590.88 cm^−1^ and the peak at 631.52 cm^−1^ can be attributed to the Fe-O bond of Fe_3_O_4_ [28,29]. Analysis of the infrared spectrum of Fe_3_O_4_@ZIF-8 shows that the peaks at 3179.34 cm^−1^ and 2926.14 cm^−1^ can correspond to the characteristic peaks of =C−O bond and C−H bond with ZIF-8 structure, respectively [30,31]. The peak of C=N bond in the imidazole ring is 1569.76 cm^−1^. The absorption peak of CN bond appeared at 1143.10 cm^−1^ and 994.82 cm^−1^, and the vibration peak of functional group in Zn-N appeared at 423.37 cm^−1^. At 591.03 cm^−1^ and 628.44 cm^−1^ the peaks appearing at −1 can be assigned to Fe-O peaks [32,33,34]. The analysis of Fe_3_O_4_@ZIF-8 by FTIR can prove that the experiment successfully complexes Fe_3_O_4_ with ZIF-8. Comparing the infrared spectrum of Fe_3_O_4_@ZIF-8 after arsenic adsorption, 3359.15 cm^−1^ corresponds to the stretching vibration region of O−H and N−H. A new strong band is observed near 427.90 cm^−1^, which can be attributed to the Zn-O-As vibration, which means the formation of a new inner spherical complex [35,36,37]. A new peak appears at 828.21 cm^−1^, which may be due to the formation of As-O groups [38] showing that arsenic was bound to Fe_3_O_4_@ZIF-8.

#### 3.2.6. XPS Analysis

Through XPS analysis of Fe_3_O_4_@ZIF-8 before and after arsenic adsorption in Figure 11, it can be found that the full spectrum in Figure 11 of Fe_3_O_4_@ZIF-8 before adsorption can detect the peaks of Zn, Fe, O, N, C, which proves to be in line with the preset Chemical composition of Fe_3_O_4_@ZIF-8 material. Fe_3_O_4_@ZIF-8 after adsorption detected the characteristic peak of As, which proved that the material successfully combined with arsenic and had an adsorption effect.

From Figure 11b, it can be observed that the positions of the peaks of Zn 2p1/2 and Zn 2p3/2, before and after adsorption, move to the position with higher binding energy, which means that the electronegativity of the surrounding atoms of Zn increases, resulting in Zn element. The binding energy also increases [39,40,41]. By fitting the peaks of As 3d after adsorption, it can be judged that after As(V) is adsorbed by Fe_3_O_4_@ZIF-8, a part of As(V) is reduced to As(III) by Fe_3_O_4_@ZIF-8.

#### 3.2.7. Surface Area Analysis (BET)

The porosity and specific surface area of the prepared Fe_3_O_4_@ZIF-8 composites were tested by N_2_ adsorption-desorption isotherms at the test temperature of 77 K. From Figure 12, it can be observed that the specific surface area of Fe_3_O_4_@ZIF-8 calculated by the BET method is 316.3593 m^2^/g, and the total pore volume measured by the single-point method is 0.224 cm^3^/g, and the micropore volume measured by the t-plot method is 0.097 cm^3^/g. The pore size is 2.83 nm, and the average mesopore size is 33.59 nm. The specific surface area of Fe_3_O_4_@ZIF-8 after adsorption of arsenic is 95.3942 m^2^/g, the measured total pore volume is 0.182 cm^3^/g, and the micropore volume is 0.017 cm^3^/g, the average pore size is 7.62 nm, and the average mesopore size is 34.35 nm. By comparison, it can be found that the specific surface area, total pore volume and micropore volume of Fe_3_O_4_@ZIF-8 after adsorption of arsenic are significantly reduced, indicating that arsenic is adsorbed to the surface of the material.

Magnetic Fe_3_O_4_ nanoparticles have been used for the treatment of arsenic from wastewater. Ref. [42] reported the maximum adsorption capacity of magnetite Fe_3_O_4_ nanoparticles occurred at pH 2, with a value of 3.70 mg/g for As (V). Due to aggregation effect, magnetic Fe_3_O_4_ nanoparticles are difficult to use in continuous flow systems [43]. Consequently, some researchers had encapsulated nanoparticles into metal organic frameworks to resolve above problem. Iron and 1,3,5-benzenetricarboxylic (Fe-BTC) used for As(V) removal from waters, and its adsorption capacity can reach 12.3 mg/g, more than 6 times that of iron oxide nanoparticles with a size of 50 nm [19]. Studies have shown that ZIF-8 demonstrated an adsorption amount of 60.03 mg/g [44] in comparison to that of MIL-53(Fe) (21.27 mg/g) [45]. Furthermore, the activated indium MOF (AUBM-1) was applied for As(V) removal from water and showed a high arsenic uptake capacity of 103.1 mg/g at a neutral pH. In this study, the maximum adsorption capacity of Fe_3_O_4_@ZIF-8 for As(V) can reach 116.114 mg/g; the result implies that the Fe_3_O_4_@ZIF-8 could be as a potential candidate for removing As(V) pollution in wastewater.

## 4. Conclusions

Characterization analysis shows that the Fe_3_O_4_@ZIF-8 materials prepared in this study have different shapes and small particle sizes, and the specific surface area can reach 316 m^2^/g. Through various characterization analyses, it can be proved that the metal-organic framework composite material was successfully prepared in the experiment; comparing the characterization results of Fe_3_O_4_@ZIF-8 before and after adsorption, it can be found that the material has a good adsorption effect on arsenic. At 25 °C, the initial concentration of arsenic is 46.916 mg/L, and the initial pH is 3. When the dosage of adsorbent is 0.4 g/L, the adsorption rate at equilibrium is 76.29%. When the dosage increases to 1.0 g/L, the adsorption rate at equilibrium increases to 99.29%. However, when the dosage of adsorbent increases to 0.6 g/L, the equilibrium adsorption rate can reach 97.20%, which is not much different from the equilibrium adsorption rate when the dosage is 1.0 g/L. Therefore, from the economic benefit to the perspective of resource saving, the optimal dosage is 0.6 g/L. When the pH value is 3, the adsorption effect of Fe_3_O_4_@ZIF-8 on arsenic can reach the best, the adsorption amount reach 59.80 mg/g, and the removal rate is 99.93%. When pH > 3, the equilibrium adsorption capacity of Fe_3_O_4_@ZIF-8 to arsenic decreases with the increase of pH, and when pH > 9.46, the decreasing rate is faster. The adsorption process of Fe_3_O_4_@ZIF-8 for arsenic has the best fit with the Langmuir adsorption isotherm equation. When the temperature is 298 K, the maximum adsorption capacity of Fe_3_O_4_@ZIF-8 for arsenic can reach 116.114 mg/g. For the adsorption kinetics, the second-order kinetic equation has the best fitting result for the adsorption kinetics of arsenic on Fe_3_O_4_@ZIF-8, with R^2^ > 0.999. By analyzing the fitting curve of the second-order kinetics, it can be found that the adsorption process of Fe_3_O_4_@ZIF-8 to arsenic-containing wastewater mainly occurs through chemical adsorption. From the analysis of thermodynamic parameters, it can be concluded that the adsorption process of Fe_3_O_4_@ZIF-8 to arsenic is a spontaneous endothermic reaction.

## Figures and Tables

**Figure 1 ijerph-19-10897-f001:**
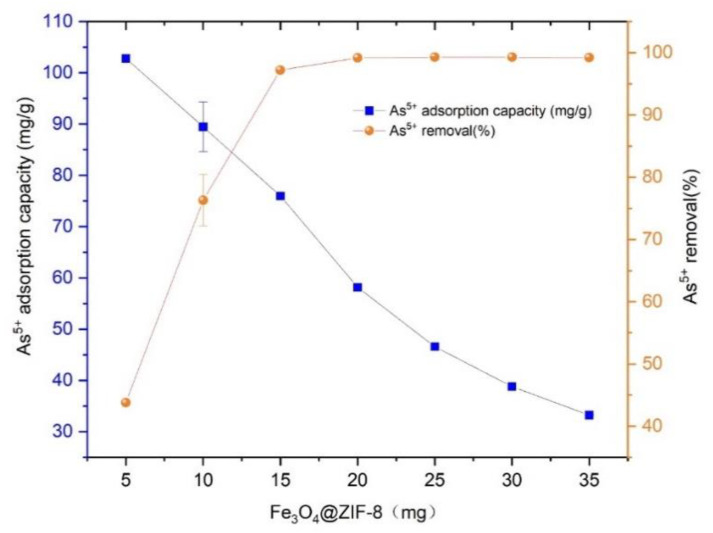
The adsorption effect of adsorbent dose.

**Figure 2 ijerph-19-10897-f002:**
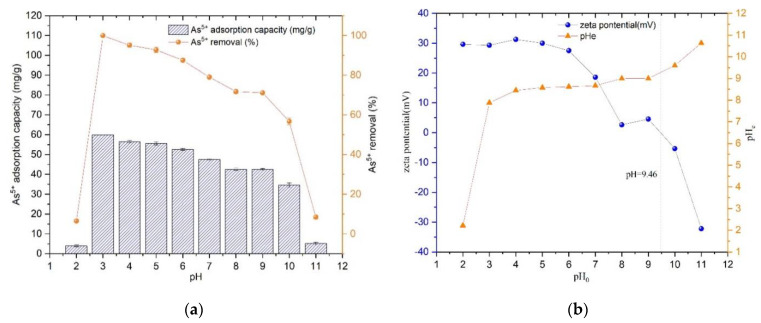
(**a**) The adsorption effect of pH; (**b**) Isoelectric point of Fe_3_O_4_@ZIF-8.

**Figure 3 ijerph-19-10897-f003:**
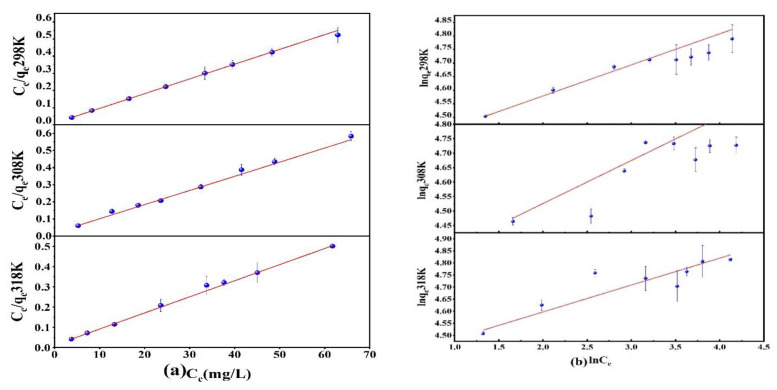
(**a**) Langmuir adsorption isotherm model; (**b**) Freundlich adsorption isotherm model.

**Figure 4 ijerph-19-10897-f004:**
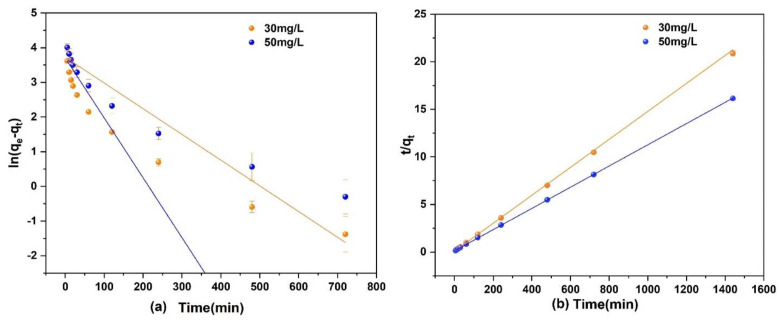
(**a**) Pseudo-first order model; (**b**) Pseudo-second order model.

**Figure 5 ijerph-19-10897-f005:**
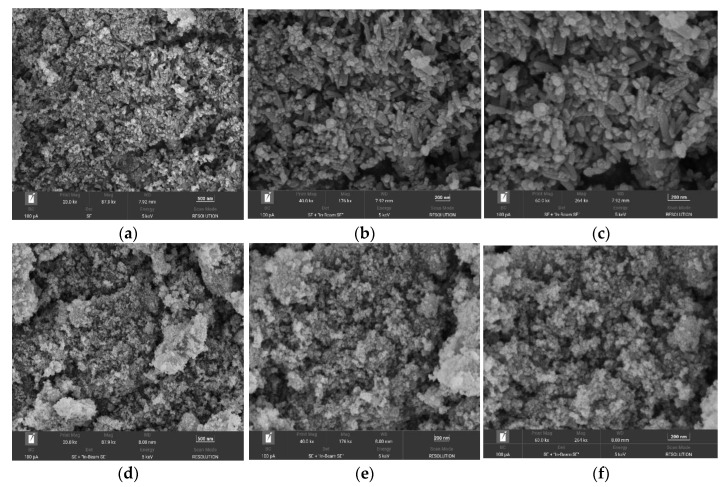
(**a**–**c**) SEM of Fe_3_O_4_@ZIF-8; (**d**–**f**) SEM of Fe_3_O_4_.

**Figure 6 ijerph-19-10897-f006:**
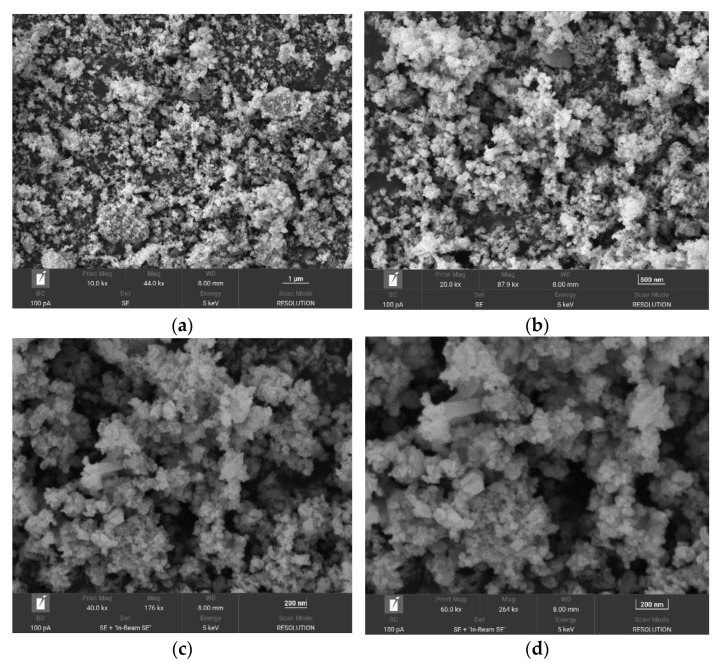
SEM of Fe_3_O_4_@ZIF-8 after adsorption. (**a**–**d**) show the effects of 10,000 times, 20,000 times, 40,000 times and 60,000 times magnification respectively.

**Figure 7 ijerph-19-10897-f007:**
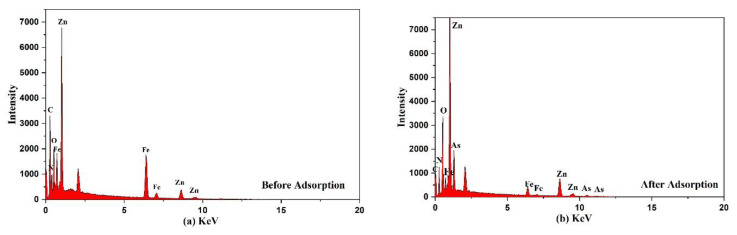
EDS spectra of the adsorbent (**a**) before (**b**) after adsorption.

**Figure 8 ijerph-19-10897-f008:**
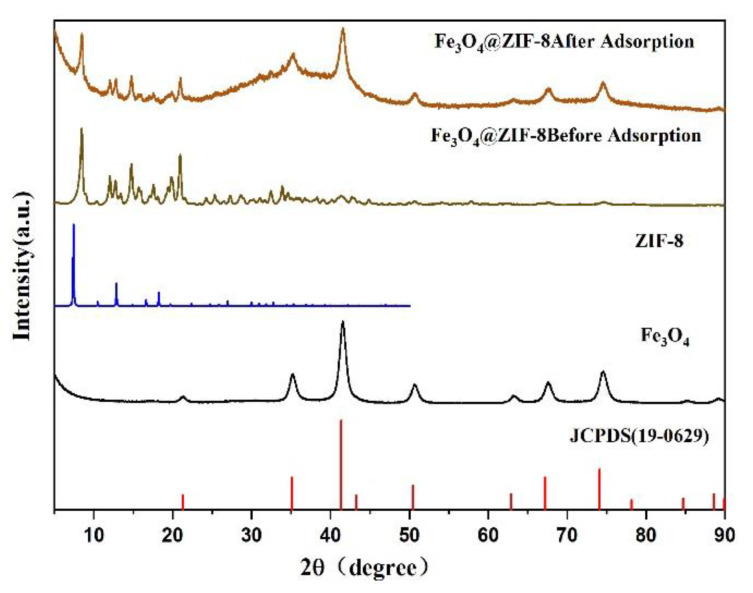
XRD pattern of synthesized Fe_3_O_4_@ZIF-8, Fe_3_O_4_.

**Figure 9 ijerph-19-10897-f009:**
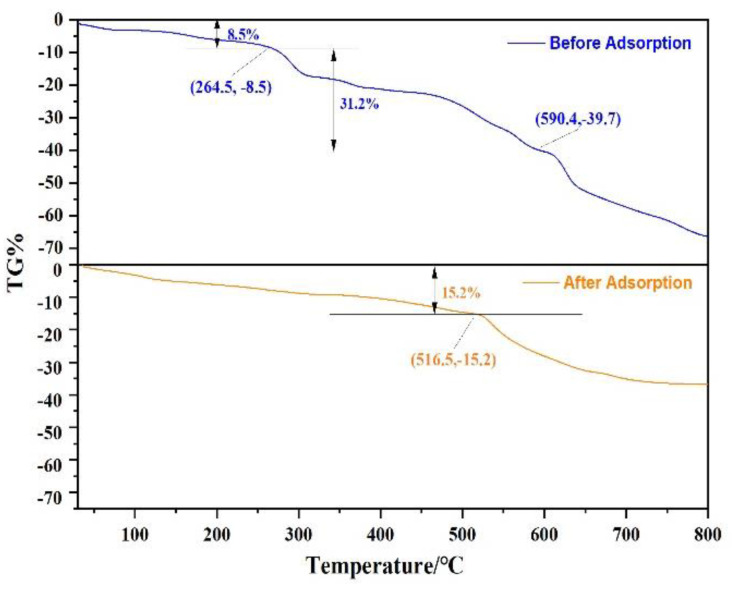
The TGA curve of Fe_3_O_4_@ZIF-8.

**Figure 10 ijerph-19-10897-f010:**
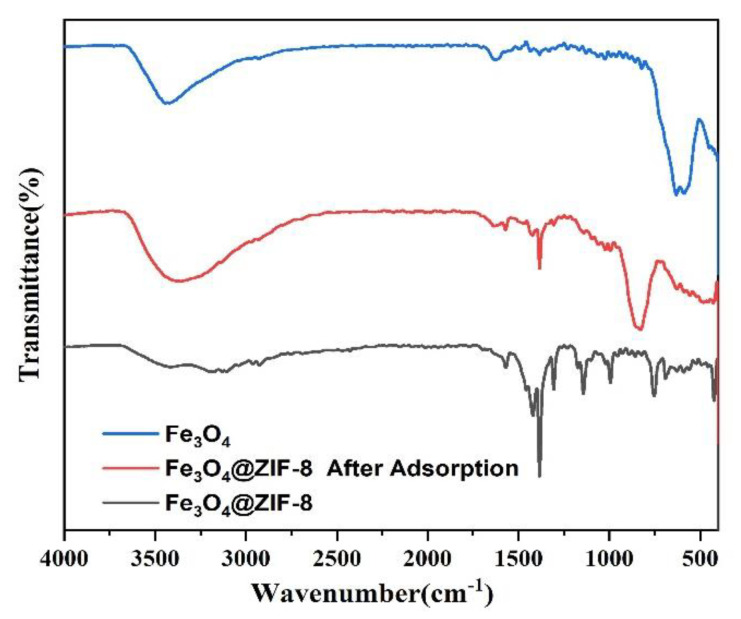
FTIR pattern of synthesized Fe_3_O_4_, Fe_3_O_4_@ZIF-8.

**Figure 11 ijerph-19-10897-f011:**
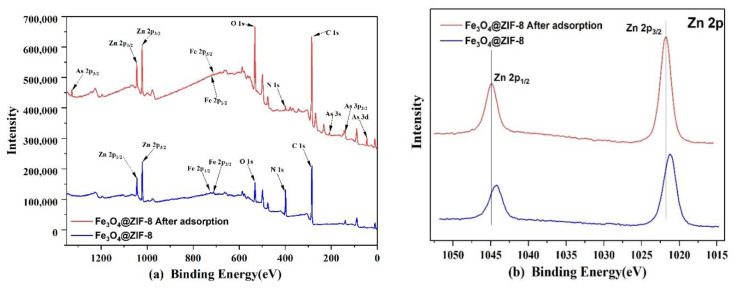
XPS spectra of Fe_3_O_4_@ZIF-8: (**a**) Wide scan; (**b**) Zn 2p core level. (**c**) As 3d core level after adsorption.

**Figure 12 ijerph-19-10897-f012:**
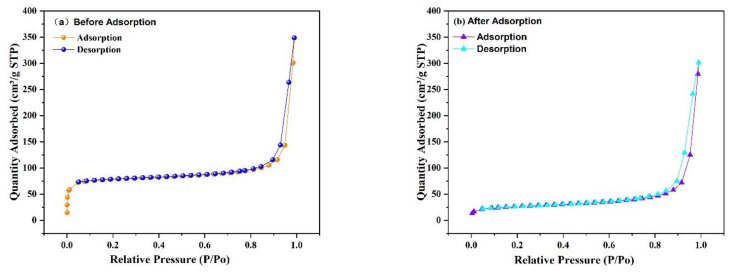
N_2_ adsorption-desorption isotherms of Fe_3_O_4_@ZIF-8.

**Table 1 ijerph-19-10897-t001:** The kinetic parameters for adsorbing As(V) by Fe_3_O_4_@ZIF-8.

Temperature	Langmuir Equation				Freundlich Equation		
(K)	q_max_ (mg/g)	K_L_ (L/mg)	R^2^	R_L_	K_f_ (L/mg)	n	R^2^
298	116.114	0.907	0.9996	0.009	77.493	8.740	0.9846
308	120.919	0.445	0.9876	0.018	68.601	6.722	0.7088
318	125.628	0.690	0.9995	0.012	79.365	8.976	0.9158

**Table 2 ijerph-19-10897-t002:** Parameters of kinetic models for As(V) onto Fe_3_O_4_@ZIF-8.

C_0_	q_e_	Quasi-First-Order Dynamics Equation			Quasi-Second-Order Dynamics Equation		
(mg/L)	(mg/g)	q_e_	K_1_	R^2^	q_e_	K_2_·10^−3^	R^2^
30	69.010	40.600	0.0173	0.9138	68.027	2.272	0.9992
50	89.186	41.558	0.0074	0.8584	89.686	0.919	0.9999

**Table 3 ijerph-19-10897-t003:** Parameters of thermodynamic for the adsorption of As(V) by Fe_3_O_4_@ZIF-8.

Temperature (K)	ΔG^0^ (kJ/moL)	ΔH^0^ (kJ/moL)	ΔS^0^ (kJ/moL)
298	−3.73	40.41	0.318
308	−4.03	40.41	0.318
318	−4.16	40.41	0.318

**Table 4 ijerph-19-10897-t004:** EDS analysis of Fe_3_O_4_@ZIF-8 before and after adsorption.

Element	Wt (%)		At (%)	
	Before	After	Before	After
C K	41.65	24.61	61.58	50.33
N K	19.39	0.75	24.58	1.32
O K	3.51	17.60	3.90	27.02
Fe K	6.58	6.38	2.09	2.81
Zn K	28.88	40.25	7.85	15.12
As K	0	10.39	0	3.41

## Data Availability

The datasets used and/or analyzed during the current study are available from the corresponding author upon reasonable request.

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
