# Peer review of "Removal of Arsenic from Wastewater by Using Nano Fe_3_O_4_/Zinc Organic Frameworks"

_ijerph, 2022, doi:10.3390/ijerph191710897_

Round 1

Reviewer 1 Report

In this paper the authors claimed the paper dealt with the catalysis (paper title) and removal of arsenic, but the full text from top to bottom has never discussed any catalysis. Also, there are no catalysts presented or described, except for the adsorption. Thus, it is not very clear what the significance and novelty of the work will be.

More questions,

1. If there was arsenic presented in a wastewater, what kind of the wastewater will be? Is there any interference?

2. Is there any correlation between trivalent arsenic and pentavalent arsenic? More discussions may be necessary for the reaction mechanism from a more in-depth perspective.

3. In lines 127 – 130, 76.3% As removal when Fe3O4-ZIF was 10mg, and 97.2% removal for 15mg, what was their molecular ratio of the adsorbents?

Reviewer 2 Report

The manuscript is well written. The title of manuscript is precise and reflected the scope of study. However, some changes/information should be added. 

1.       Throughout the text, please insert a space before brackets for citations in the text

2.       Introduction, Line 38: The authors should clarify if arsenic is a non-metallic element or if it is a metalloid, and cite at least a reference for this (e.g. https://doi.org/10.1155/2017/3037651)

3.       Introduction, Lines 48-49: Arsenic should be written with lowercase inside the sentence.

4.       Introduction, at the end of this section the authors should more clearly state the aim of the study and to highlight the novelty / originality of this

5.       Experimental section. Only the synthesis of adsorbents is presented. The authors should also present here the preparation of adsorption experiments and the analytical techniques used for materials characterization and for As(V) determination in solution. Which chemical was used for As(V) solution preparation?

6.       Experimental section. Please provide the producers (name, country, city) for the chemicals used in experiments

7.       Results and discussion. Please compare the adsorption capacities for studied adsorbents with those reported in literature for other adsorbents.
